# Forest edges are globally warmer than interiors and exceed optimal temperatures for vegetation productivity

Josephine Elena Reek [1] ✉, Thomas W. Crowther [1], Thomas Lauber [1], Sebastian Schemm [2], David Parastatidis[3], Nektarios Chrysoulakis [3], Mengtian Huang [4], Shilong Piao [5], Constantin M. Zohner [1] & Gabriel Reuben Smith [1]

Forests not only regulate the global climate by absorbing carbon dioxide but also shape local biophysical conditions by creating microclimates that buffer temperature extremes. However, ongoing deforestation and fragmentation are transforming forest interiors into edge environments, which may differ markedly in their microclimatic conditions and undermine local climate-regulating functions. Here, we quantify how proximity to forest edges alters thermal conditions across biomes and seasons using global satellite-derived surface temperature data from nearly 13 million sites. We find that forest edges are consistently warmer on average than interiors, with the magnitude of warming varying with biome type and season. During summer months, surface temperature at edges frequently exceeds the optimal temperature for vegetation productivity, particularly in tropical forests. These results suggest that continued loss of interior forest will reduce the capacity of remnant forests to buffer local climate conditions, potentially hampering ecosystem productivity and resilience.

Forest ecosystems are home to over 50% of terrestrial biodiversity[1] and play a central role in climate change mitigation through carbon storage[2,3]. Forests also shape local temperatures through their biophysical effects, including shading and transpirative cooling during photosynthesis. These cooling effects not only influence key temperature-sensitive ecosystem processes[4–7] but also enhance human wellbeing and support livelihoods[8]. Yet, these climate benefits of forests are increasingly impaired by deforestation and environmental disturbances[9,10], with an estimated 129 million hectares of forest lost between 1990 and 2015[11].

Deforestation not only reduces forest area but also increases fragmentation, breaking up large, contiguous forests into smaller patches with a greater proportion of edge area. In fact, 70% of forests worldwide lay within 1000 m of an edge in 2015[2,12]. But while we know that forest removal directly alters local temperatures[13–17], the temperature consequences of forest fragmentation remain less clear[18]. Prior research suggests that conditions at edges may differ significantly from a forest's interior[18–21], with field studies observing weaker sub-canopy cooling effects and lower temperature buffering ability near edges[22–26]. Similarly, a remote sensing study has found higher daytime temperatures in tropical edge forests compared to intact

interiors[27]. However, temperature dynamics at the landscape scale, where multiple patches and land cover types interact, can differ from local-scale patterns[18,28]. For instance, Mendes & Prevedello (2020)[28] found that when considering the entire landscape, temperatures tended to be cooler overall in more fragmented areas. This may be due to "vegetation breezes", where air circulation is enhanced over heterogeneous land cover, affecting rainfall patterns and evapotranspiration[18,20,28]. These complexities underscore the need to distinguish between temperature changes within forest patches and those across the whole landscape. While fragmentation changes the amount of edge areas, this study is specifically looking at edge effects – the microclimatic differences between forest edges and interiors—which cannot be directly extrapolated to explain temperature patterns across entire fragmented landscapes.

Research suggests that the cooling effect of forests may get stronger under higher macroclimatic temperatures[15,29]. This has important implications under global warming patterns. Understanding the temperature dependence of microclimatic forest cooling is relevant not only for local human populations, but also for temperature-sensitive ecosystem processes. One such process is ecosystem productivity, whose thermal optimum varies

[1]Institute for Integrative Biology, ETH Zurich, Zürich, Switzerland. [2]Institute for Atmosphere and Climate, ETH Zurich, Zürich, Switzerland. [3]Foundation for Research and Technology Hellas (FORTH), Institute of Applied and Computational Mathematics, Remote Sensing Lab, Heraklion, Greece. [4]State Key Laboratory of Severe Weather and Institute of Global Change and Polar Meteorology, Chinese Academy of Meteorological Sciences, Beijing, China. [5]Institute of Carbon Neutrality, Sino-French Institute for Earth System Science, College of Urban and Environmental Sciences, Peking University, Beijing, China. ✉e-mail: josephine.reek@alumni.ethz.ch

geographically, typically lying above current temperatures in boreal regions and below them in tropical regions[30]. Microclimatic edge effects, where temperatures near forest edges differ from interiors, could shift the local thermal environment that organisms experience, adding another dimension to the mismatch between current and optimal productivity conditions. Consequently, proximity to the productivity optimum may depend not only on geographic location, but also on position relative to the forest edge and, by extension, on the amount of edges in a landscape. Thus, a forest in a given location might experience optimal or sub-optimal temperatures for productivity depending on whether it forms part of a large, intact patch or lies close to an edge, which is more likely in fragmented landscapes.

Here, we address these questions at a global scale by examining satellite-sensed surface temperature ($T_{surf}$; for forests, this represents the surface temperature of leaves at the top of the canopy) across nearly 13 million sites, encompassing forest-adjacent, forest edge, and forest interior areas. We quantify the relationship between forest $T_{surf}$ and distance-from-edge across seasons and biomes. Finally, we compare the observed differences to published empirical estimates of $T_{surf}$ optima for ecosystem productivity[30]. In doing so, we test three hypotheses: **H1**) $T_{surf}$ near forest edges will more closely match $T_{surf}$ outside of forests than interior forest $T_{surf}$, reflecting reduced microclimatic buffering at the edges; **H2**) The magnitude of the $T_{surf}$ edge effect will increase with macroclimatic temperature, indicating that edge effects intensify under hotter conditions; and **H3**) in tropical forests, interior $T_{surf}$ will align more closely with ecosystem productivity optima, whereas in higher latitude forests, edge temperatures may be more favourable for productivity due to cooler baseline temperatures.

Our findings confirm that forest edge temperatures more closely match non-forest temperatures, both being generally higher than interior temperatures. Additionally, the edge effect positively correlates with macroclimatic temperature, amplifying in hotter regions. However, contrary to **H3**, elevated edge temperatures do not translate into improved conditions for ecosystem productivity at the edges, as these temperatures already exceed optimal productivity thresholds.

## Results and discussion
### Characterising $T_{surf}$ around forest edges
While previous studies have shown that subcanopy temperatures exhibit a buffering effect against macroclimatic warming[16,31,32], we find that on average, forest $T_{surf}$ is uniformly colder than that of surrounding areas (Methods; Fig. 1a). This consistent cooling effect may be partly driven by transpirative cooling, a well-established mechanism that reduces surface temperatures[15]. Across all biomes and seasons, we observe a distinct edge effect, with $T_{surf}$ gradually increasing from forest interiors to edges and adjacent non-forest areas. Notably, commonly used satellite- derived $T_{surf}$ measurements at spatial resolutions of 250 m or 1000 m cannot capture this effect, as each pixel would encompass entire edge zones. By employing a 30 m resolution, we can delineate multiple pixels from edge to interior, allowing us to characterise the temperature curve more precisely.

In support of **H1**, we find that forest edge $T_{surf}$ in most biomes and seasons are warmer (i.e. their $T_{surf}$ is more similar to the surrounding non-forest) than forest interiors are, with the exception of boreal winter (Fig. 1d). This exception may stem from low transpirative cooling in colder conditions, and forests' lower albedo compared to surrounding snow. Light snow has a high albedo, reflecting incoming radiation and thereby cooling the landscape compared to darker forests, which absorb more radiation. In both boreal and temperate regions, the edge-interior $T_{surf}$ difference is more pronounced in summer than winter. Thus, consistent with **H2**, the edge effect intensifies with increasing temperature, aligning with previous findings[15,29].

To further investigate whether the temperature difference between edge and interior forest depends on regional macroclimate, we calculated $^{\Delta T}/_{\Delta D}$, the change in $T_{surf}$ with change in $\log_{10}$-transformed distance from forest edge ($\Delta D$), for each satellite scene (Methods). Then, we used inverse variance weighted regression to compare our $^{\Delta T}/_{\Delta D}$ estimates with the local

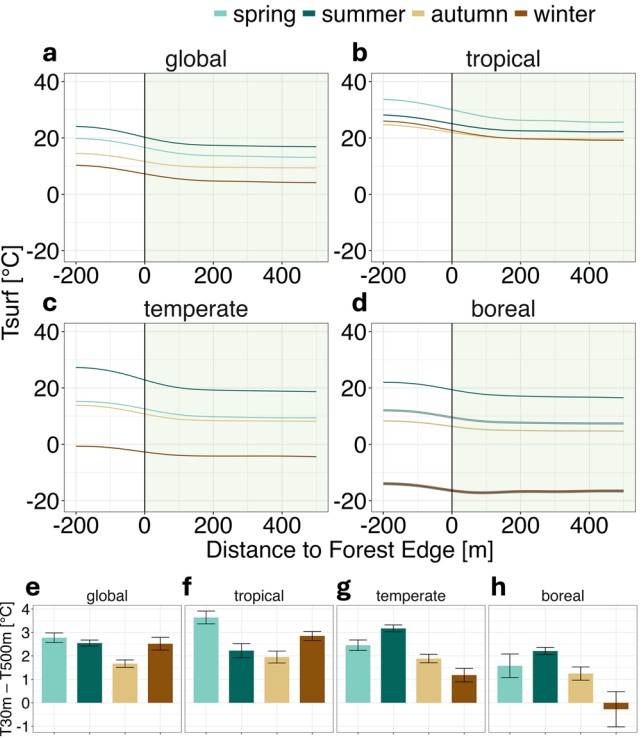

**Fig. 1 | Surface Temperature ($T_{surf}$) around forest edges.** corrected for satellite scene and overpass, as well as elevation: y-axis represents partial residuals. x-axis represents distance to forest edge in metres with 0 being the edge, positive numbers indicating the distance into the forest (green shading), and negative numbers the distance outside the forest; dark blue - summer, light blue - spring, beige - autumn, brown – winter, grey – standard error. **a–d** curves of $T_{surf}$ across forest edges; **e–h** difference in $T_{surf}$ between the edge (30 m) and the forest interior (500 m), error bars denote standard error; Curves were predicted based on BAMs with: global $n = 11,064,798$; tropical $n = 3,314,823$; temperate $n = 5,086,080$; boreal $n = 2,237,755$; Statistics for the underlying BAMs are documented in Supplementary Table 3. For delineation criteria of seasons, see Supplementary Table 1, for delineation of biomes, see Supplementary Table 2. For the same analysis in the tropics using wet and dry season, see Supplementary Fig. 2. For the difference in $T_{surf}$ between the edge (50 m) and the forest interior (1000 m) (analogous to **e−h**) see Supplementary Fig. 9.

macroclimate, calculated as the average temperature (either $T_{surf}$ or modelled mean annual temperature; Supplementary Figs. 4 and 5) of the satellite scene in question. In further support of **H2**, we found that $^{\Delta T}/_{\Delta D}$ is positively correlated with macroclimatic temperature (Fig. 2), indicating that the cooling effect from edge to interior strengthens under warmer macroclimatic conditions. However, the $R^2$ value of 0.20 suggests that macroclimatic temperature explains only a modest fraction of the observed variability. Additional factors identified in previous studies[18], such as edge orientation, age, surrounding land cover, and forest type, likely contribute to the variability in the edge effect. Moreover, the origin and anthropogenic modifications of forest edges may influence temperature dynamics, presenting a valuable avenue for future research. Nevertheless, the findings presented here suggest that the cooling capacity of forest interiors may increase under rising temperatures and that this adaptive cooling capacity is weaker at edges.

Previous studies have demonstrated that $T_{surf}$ is strongly influenced by both albedo and evapotranspiration[15]. Darker forest canopies typically exhibit lower albedo than surrounding non-forest areas, leading to relatively warmer surfaces, particularly under snow cover[15,29]. Conversely, evapotranspiration, which is usually higher in forests, has a cooling effect[15]. This cooling effect is most pronounced under warmer, wetter conditions, especially during the summer growing season[33,34]. Even under drier conditions, forests can maintain higher evapotranspiration rates—and thus cooling

**Fig. 2 | Relationship between the temperature gradient from edge to interior and macroclimatic T_surf. a** Full dataset including all sampled locations (*n* = 39,455). **b** Subset of locations inside the forest and where cooling increased towards the forest interior (*n* = 28,141). Each point represents the strength of the temperature edge effect in one satellite scene, calculated as the change in T_surf (ΔT) with log-transformed distance to forest edge (Methods). Points are colour-coded by density (blue = high density, brown = low density), with larger points indicating greater weight in the inverse variance-weighted quadratic regression. The black curve represents the fitted regression line, with shaded areas denoting confidence intervals. In both panels, the positive quadratic trend indicates a stronger cooling effect from edge to interior under higher macroclimatic temperatures. $R^2$ values indicate that macroclimatic temperature explains 19.7% of the variation in (**a**) and 16.3% in (**b**). All regression coefficients (intercept, T_surf, and quadratic term) are statistically significant (*p* < 0.001).For seasonal analyses, see Supplementary Fig. 3; for analyses using MODIS-derived T_surf and annual mean temperature, see Supplementary Figs. 4 and 5.

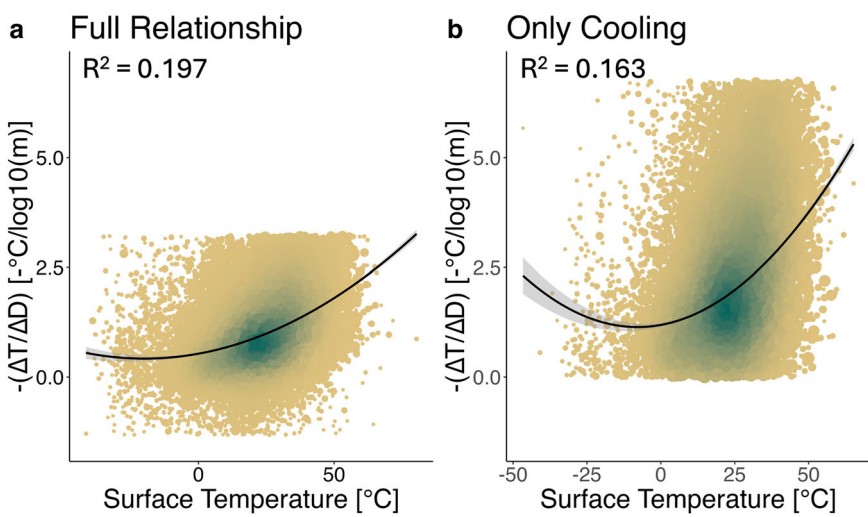

effects—due to deeper root systems that access water more effectively than surrounding grasslands[35,36]. Thus, while our study primarily documents observed temperature patterns rather than underlying mechanisms, the pattern of stronger edge-interior temperature differences at higher macroclimatic temperatures aligns with the dominance of evapotranspiration-driven cooling under warm conditions and albedo-driven warming under cooler conditions.

## Impacts of T_surf edge effects on forest productivity

T_surf measurements represent the surface temperature of leaves at the top of the canopy (or, at low canopy cover, a combination of canopy and ground temperatures), a key determinant of photosynthesis[37]. As such, the observed T_surf differences between edges and interior forests are expected to influence ecosystem productivity, a key driver of C fluxes at the global scale[2]. Because forest fragmentation transforms interior forest into edge habitat, understanding how T_surf differences between edges and interiors impact productivity can clarify the knock-on effects of fragmentation on the C cycle within remnant forests.

Using published empirical estimates of the T_surf optimum for ecosystem productivity[30], we assessed whether the observed summer T_surf at forest interiors or edges was closer to the expected T_surf optimum in each location. This comparison is feasible because both datasets use satellite-derived T_surf (see methods). Contrary to **H3**, interior T_surf during the summer was consistently closer to the productivity optimum across all biomes (Fig. 3). In the tropics, both interior and edge T_surf exceeded the optimal temperature for productivity, in line with previous findings based on air temperature[30]. Ecosystem-level productivity is thus expected to be lower than it would under cooler, optimal temperature conditions, especially in the hotter edge areas. In temperate and boreal regions, interior T_surf was typically close to the productivity optimum (<1 °C difference on average), while edge T_surf remained higher and less favourable for productivity. Thus, contrary to the expectation that higher temperatures at edges in cooler biomes might enhance productivity, we instead observed a globally pervasive negative effect: On average, summer T_surf at forest edges was consistently less conducive to optimal productivity than interior T_surf across all biomes.

Temperature is one of many factors influencing productivity and biomass[38–40]. Our findings align with studies reporting reduced biomass at tropical forest edges[19,41,42], where we observed the largest

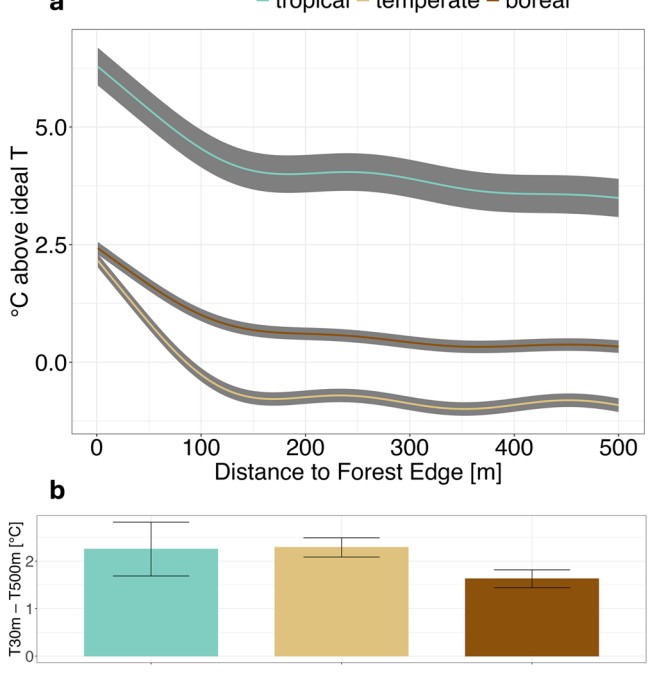

**Fig. 3 | Average difference between observed T_surf and ideal T_surf for ecosystem level productivity during the summer.** Blue – tropical, beige – temperate, brown – boreal. Standard error shown in grey/ errorbars; **a** °C above ideal temperature for productivity with distance to forest edge; **b** difference of °C above ideal at forest edge (30 m) and forest interior (500 m). Predicted lines based on BAMs with tropical (*n* = 101,803) temperate (*n* = 451,074), boreal (*n* = 494,469). Statistics for the BAMs are documented in Supplementary Table 4. For the same analysis in tropical regions using wet season, see Supplementary Fig. 6. For the same analysis in evergreen forests over the full year, see Supplementary Fig. 7. For the difference in T_surf between the edge (50 m) and the forest interior (1000 m) (analogous to **b**) see Supplementary Fig. 11. For delineation of seasons, see Supplementary Table 1, for delineation of biomes, see Supplementary Table 2.

deviations from ideal temperature for productivity. In temperate and boreal forests, however, biomass responses to edge conditions are more variable, with studies documenting both increases and decreases in biomass[19,43,44]. This does not directly match our results, as we observed more favourable temperatures for productivity in forest interiors across all biomes. However, the deviation from the temperature optimum was smaller for those biomes, and the negative temperature effects may have been outweighed by other factors like increased light availability. Additionally, one study that documented increased biomass at temperate forest edges also found that edges were more vulnerable to growing season heat stress than interiors, with growth declining three times faster at edges than interiors as a response[44]. This aligns with our observations that current temperatures at edges may already be approaching levels that inhibit productivity. These findings also qualitatively support a recent study showing that, under current conditions, tropical trees already experience temperatures high enough to cause leaf death[45]. This study estimated that tropical forests can withstand air temperature warming of $3.9 \pm 0.5\,°C$ before reaching a tipping point for metabolic function at the ecosystem level.

Ultimately, the effect of heightened edge temperatures on productivity also depends on the potential for thermal adaptation. Our analysis relies on a spatially explicit $T_{opt}$ dataset but does not account for potential differences between forest edges and interiors. A recent study suggests that $T_{opt}$ has increased with rising macroclimatic temperatures over recent decades[46]. If such adaptive mechanisms can operate over short spatial and temporal scales, forest edge vegetation may adjust to higher temperatures by raising its $T_{opt}$, potentially reducing the divergence between edge temperatures and optimal productivity conditions. Nevertheless, our finding that observed edge temperatures already exceed the optimal range for productivity in many regions underscores the importance of accounting for edge effects in assessments of forest resilience and carbon cycling under ongoing climate change and fragmentation. Future studies should examine whether $T_{opt}$ can adjust at the microclimatic scale and assess how such adaptations might mediate the thermal stress associated with increased edge exposure.

## Conclusions

Global temperatures continue to rise[47], hindering forest productivity[30]. At the same time, forest fragmentation is positioning more of the world's remaining forests in proximity to edges[2,12], where forest cooling capacity is weakened (Fig. 1). Our work suggests that these dual global change pressures may interact to drive forest conditions further away from the productivity optimum, especially in tropical forests. Moreover, since transpiration associated with photosynthesis contributes to forest cooling[15], productivity losses may further increase forest $T_{surf}$ and thereby constrain growth. Therefore, increasing the proportion of forests exposed to edge effects may not only elevate local surface temperatures but also constrain temperature-sensitive processes like photosynthesis, potentially impacting forest carbon dynamics and ecosystem resilience.

## Methods

We combined 30 m resolution satellite sensed surface temperature ($T_{surf}$) data from Parastatidis et al.[48] with a dataset of distance to forest edge. Data was retrieved on a global scale for the full year 2010 (every 16 days), sampling points within 1000 m of forest edges (both sides of the edge) for a total of 12,969,118 points. We used generalised additive models to investigate how the pattern of $T_{surf}$ around forest edges varies globally and by season. In order to investigate potential effects on productivity, we compared our observed $T_{surf}$ to published estimates of the ideal $T_{surf}$ for optimal productivity during growing seasons[30].

### Surface temperature data

We used a 30 m resolution geospatial $T_{surf}$ data product based on measurements from Landsat satellite 5[48]. The year 2010 was used for this analysis because it allowed us to use the Landsat-derived $T_{surf}$ data[48] as well as high

quality and high resolution forest cover data[49]. Commonly used satellite-derived $T_{surf}$ measurements at spatial resolutions of 250 m or 1000 m cannot characterise the edge effect we are investigating, as the entire forest edge would lie within one or two pixels. The 30 m resolution of this dataset fits several pixels between the forest edge and the point where $T_{surf}$ stabilises in the forest interior (or exterior), allowing for characterisation of the curve. The $T_{surf}$ dataset uses a single channel algorithm and emissivity based on NDVI (also measured by Landsat) and has a reported error of 1.40 °C (bias 0.31). It is based on Landsat collection 1. A more detailed description of the dataset can be found in the original publication[48]. Satellite-sensed $T_{surf}$ is commonly used to quantify local forest temperature effects[13,50]. Since it measures the surface temperature of an object instead of the surrounding air, it allows us to compare non-forest points at ground level with forest points at canopy height[51]. Sampled points are shown in Supplementary Fig. 1.

The satellite data consists of discrete "scenes", which each cover an area of ca. 185 km × 180 km[52]. Each scene is captured once per satellite overpass, covering the entire globe every 16 days; we refer to each unique combination of a particular satellite scene and a particular overpass as an "SxO".

Within each SxO, we randomly sampled 1000 points using the Python API for Google Earth Engine[53]. We filtered out points with cloud cover, which prevents $T_{surf}$ measurement, and points that were not on land. Then, using the forest cover map produced by Hansen et al.[49], we classified our points as either falling within forest (≥30% canopy cover) or outside of forest (<30% canopy cover). For each point falling within a forest, we calculated the Euclidian distance to the nearest non-forest pixel (30 m resolution), and for each point falling outside of a forest, we calculated the Euclidian distance to the nearest forest pixel. Since we were interested specifically in temperature variation near forest edges, we then removed points with a distance of more than 1000 m to the nearest forest edge. According to this procedure, we collected sample points for the full year of 2010, resulting in a global dataset across 23 overpasses and 6277 scenes containing a total of 12,969,118 points. The points were grouped into four seasons in whole overpasses (Supplementary Table 1).

As cloud cover prevents $T_{surf}$ measurement, our analyses specifically concern only cloud-free days, which tend to exhibit larger temperature differences in forest vs. non-forest areas than cloudy ones[23,31]. The satellite always passes the equator between 10:00 am and 10:25 am local time[54], and the orbit is designed to keep all measurements close to this in local times. Studies on diurnal variation indicate that differences in temperature between forests and surroundings are small or even reversed at night, and rise during daytime[15,23,51,55]. This means that absolute temperature differences between forest and non-forest partially depend on how long after sunrise the measurement was performed. Supplementary Fig. 8 shows the distribution of scene capture time in hours after sunrise for our study. 59% of our SxOs were captured between 3.5 and 5.5 h after sunrise, with 94%, 55%, and 19% for the tropical, temperate, and boreal biomes respectively. A part of the temperate and the majority of boreal scenes were captured later with reference to sunrise, mainly because sunrise happens earlier during the summer. Indeed, in high latitude boreal regions a polar night occurs during the winter, where the sun does not rise, in which case we cannot obtain $T_{surf}$ measurements. Conversely, the sun does not set during the summer. The lead-up to this period explains why some of our boreal scenes were captured up to 12 h after sunrise. 9% of the SxOs in the boreal fell into high latitudes and summer, where the sun does not set at all and thus sunrise is not applicable for the purpose of our investigation and could not be computed.

Based on studies investigating the diurnal cycle during the growing season, the maximum offset between forest and non-forest temperatures is observed around midday[23,51,55]. We expect the later time of image capture in the boreal biome to lead to relatively larger offsets between forest and non-forest. Since our overall conclusion shows smaller offsets in the boreal compared to other biomes, but our boreal estimates are expected to be relatively high, is unlikely to affect our overall conclusion.

Our study year of 2010 was an El Niño year, with relatively high temperatures[56]. However, in recent years mean annual temperature has

been consistently higher than it was in 2010[57]. Our findings that observed temperatures exceed the optimum for ecosystem level productivity (Fig. 3) are thus likely conservative under current conditions.

Cloud cover is known to affect size but not direction of estimates and affects all biomes, and the time of measurements (time of day, as well as year) leads to conservative estimates with respect to our conclusion. Thus, overall trends and patterns are unlikely to be affected by either of these variables.

## Data analysis and $T_{surf}$ patterns

All subsequent analyses were performed in R studio[58] using R packages tidyverse[59] and ggplot2[60]. We defined ecoregions based on Dinerstein et al.[61] and sorted the ecoregions into biomes (Supplementary Table 2).

For the characterisation of $T_{surf}$ around forest edges (Fig. 1, Supplementary Fig. 2), datapoints were filtered further to a distance of max. 500 m (including 500 m) to the nearest forest edge. Distances outside the forest have negative values, inside the forest positive.

As elevation is related to temperature, and, sometimes, edge formation, we control for it in our analysis. For that purpose we computed the mean of two elevation datasets that are each at 30 m resolution (refs. 62,63). For some points along coastlines this led to NA values, as they are not considered to lie on land across all datasets, which we excluded (<0.1% of datapoints that had $T_{surf}$ and Biome information). Since each SxO represents a different day and thus different abiotic conditions, it is important to compare points only within each individual SxO, rather than across multiple SxO. At the same time, our overall goal for each biome and season was to measure the mean relationship between forest edge distance and $T_{surf}$. Accordingly, we used generalised additive models for very large datasets (BAMs) as implemented in the R package mgcv[64] with $T_{surf}$ as the response variable, a fixed smoothed term for distance from forest edge as well as elevation, and a smoothed additive random effect for the unique SxO (statistics for all subsequently described models of this type are documented in Supplementary Tables 3–7). BAMs were run with discrete=True. Then we used the "predict" function, excluding SxO and elevation. We predicted a value for each metre from −200 (outside the forest) to 500 (inside the forest) with elevation set to its mean and SxO to the most frequent level. This allowed us to calculate on partial residuals that leave only the effect of distance to forest edge on $T_{surf}$ and exclude the effect of SxO and elevation.

The random effect controls for all $T_{surf}$-affecting variation (abiotic and otherwise) that makes a focal SxO different from another SxO, except for $T_{surf}$ variation attributable to distance from forest edge. In this way, the model fits the slope that best explains the relationship of $T_{surf}$ to forest edge distance while controlling for all large-scale environmental variation across the complete set of Landsat scenes. We ran models within each unique combination of biome and season, all following this same approach. Statistics for the BAMs are presented in Supplementary Tables 3–7.

In order to compare temperature differences between points at the forest edge (30 m) and points in the interior (500 m) we used the same BAMs as described above to predict the temperature in each location. Thus, the edge temperature for these comparisons is the temperature 30 m from the edge into the forest as predicted by our BAM, fitted on observed measurements across different edge distances and accounting for SxO as well as elevation. We extracted the standard error of those predictions and computed the standard error (SE) of the difference assuming independence using:

$$SE(T_{edge} - T_{interior}) = sqrt\left(SE(T_{edge}) + SE(T_{interior})\right)$$

We placed the forest edge prediction at 30 m to get as close to the edge as possible, while avoiding influence of non-forest area on our 30 m resolution. 500 m was chosen for the inside since temperature curves had flattened out by then (Fig. 1). Additionally, Supplementary Fig. 9 shows the same analysis comparing 50 m and 1000 m (statistics for the BAMs in Supplementary Table 9).

In order to ensure that our SxO correction is sufficient, and the results are not based on autocorrelation, we calculated a Moran's i on the residuals of all BAMs (using a random subset of 75,000 points, Supplementary Table 8). While the values were significant, they are very low (0.0268 for boreal winter and <0.01 for all other models) and we do not expect our conclusions to be influenced by it.

Previous studies indicate that forest microclimates have cooler temperatures relative to their surroundings in hotter conditions[15,29]. Similarly, we sought to determine whether the relationship between forest edge distance and $T_{surf}$ was dependent on regional macroclimate. To do this, we first used linear regression to measure the relationship between $\log_{10}$-transformed distance to forest edge and $T_{surf}$ and elevation, running a model for each SxO individually. These models use datapoints from the forest edge up to 1000 m inside the forest and use only SxO with at least 4 observations. The beta coefficients of these models describe the $T_{surf}$ change per distance change, $^{\Delta T}/_{\Delta D}$, per scene. In other words, they describe the slope of $T_{surf}$ per $\log_{10}$-transformed distance to forest edge. To avoid outliers, we applied a Hampel filter which considers as outliers points that fall outside the interval of median ±3 mean absolute deviations. We also provide the same analysis without any outlier filter in Supplementary Fig. 10.

We then performed a second, quadratic regression, modelling $^{\Delta T}/_{\Delta D}$ as a function of each SxO's average $T_{surf}$, which was calculated as the arithmetic mean value of all the points sampled in that scene. To propagate the error in $^{\Delta T}/_{\Delta D}$ across to this second analysis, we weighted each $^{\Delta T}/_{\Delta D}$ by the inverse of its variance, as is commonly done in meta-analysis[65] (Fig. 2). To check the robustness of this result, we performed additional analyses using only points within the forest and in which the forest interior had lower temperatures than the edge, but comparing to the mean temperature of points sampled in the full scene (inside and outside the forest) as before. We also tried using MODIS-derived 1000 m resolution $T_{surf}$[66], and compared Landsat-derived temperature edge curves to mean annual temperature[67] instead of the mean temperature of the scene (Supplementary Fig. 4; Supplementary Fig. 5). The increase in edge effect with increasing macroclimatic temperatures held across these analyses.

## Effects on productivity

Photosynthesis is temperature-sensitive[30,68,69]. The observed differences in $T_{surf}$ between forest interiors and edges are thus likely to influence plant productivity. To estimate this potential influence, we paired our measurements with a previously-published map of $T_{surf}$ optima for ecosystem productivity (see ref. 30, Supplementary Fig. 22) at a resolution of 1/12°x1/12°, subsequently abbreviated as $T_{opt}$.

This $T_{opt}$ dataset uses land surface temperature from MODIS (MYD11A2 version 6[70]) combined with $NIR_v$ (near infrared reflectance) from MODIS NIR reflectance and MODIS NDVI according to ref. 71. $T_{opt}$ could thus be calculated using data from 2003 to 2013, which includes our study year 2010. A more detailed description of the dataset can be found in the original publication[30]. The overpass time of the MODIS Aqua satellite (used for the reference land surface temperature in the $T_{opt}$ dataset) is ca. 13:30, and thus somewhat later than that of Landsat (used for our observed $T_{surf}$), which passes the equator between 10:00 am and 10:25 (see details above). Based on diurnal temperature patterns[15,23,51,55], MODIS likely measures a slightly higher temperature for the same day and place compared to Landsat. The day with the highest productivity thus likely registered a slightly higher temperature on MODIS than Landsat. The optimal temperature above which productivity declines is then higher and thus less often reached in Landsat measurements. As our observations still register higher than optimal temperatures, our estimates are likely conservative in how often these higher than optimal temperatures are reached and how far they are transgressed.

For this analysis, we focussed on summer $T_{surf}$ (defined according to hemisphere-specific dates listed in Supplementary Table 1) in order to capture the period of maximum vegetation activity and used datapoints up to 1000 m into the forest. Since tropical regions are often characterised by wet/dry seasonality[72], we performed a supplementary tropical analysis

focussing on wet season $T_{surf}$ (Supplementary Fig. 6) as well as an analysis using the full year in evergreen forests (Supplementary Fig. 7). As we are interested in forest productivity, points outside the forest were excluded for all analyses.

Values of $T_{opt}$ were extracted from Huang et al.[30] for the same coordinate points as the $T_{surf}$ data used for our previous analyses. Then, the difference between observed $T_{surf}$ and $T_{opt}$ was computed for each point. To be included in the analysis, a satellite scene needed at least 500 observations from at least two different overpasses as well as at least one observation 800 m or further into the forest. Then, within each biome, we ran a BAM model implemented in the R package mgcv[64] with the difference between observed $T_{surf}$ and $T_{opt}$ as the response variable, and three predictors: a fixed smoothed term for distance from forest edge and elevation, and a smoothed additive random effect for the unique SxO. We used the "predict" function, excluding SxO and elevation (predicting for every metre from 0 to 500, elevation set to its mean, SxO set to the most frequent), leaving only the effect of distance to forest edge on $T_{surf}$-$T_{opt}$. BAMs were run with discrete=True (except for Supplementary Fig. 7, which was computed using discrete=False). Statistics for the BAMs can be found in Supplementary Tables 4, 6, 7. We compared the difference to the optimal temperature between points at the forest edge (30 m) and points in the interior (500 m) using the same BAMs as described above to predict the temperature in each location, analogous to Fig. 1. Additionally, Supplementary Fig. 11 shows the same analysis comparing 50 m and 1000 m (statistics for the BAMs in Supplementary Table 4).

### Reporting summary
Further information on research design is available in the Nature Portfolio Reporting Summary linked to this article.

## Data availability
Data can be found under https://doi.org/10.5281/zenodo.15743880.

## Code availability
Code can be found under https://doi.org/10.5281/zenodo.15743880.

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

## Acknowledgements
The authors thank Lukas Graz and Krzysztof Cybulski for statistical consulting. J.E.R., C.M.Z. and T.W.C. were supported by DOB Ecology and the Bernina Initiative. G.R.S. was funded by the Bernina Initiative and Ambizione grant (PZ00P3_216194) from the Swiss National Science Foundation. D.P. and N.C. were supported by an *urbisphere* ERC-Synergy Grant (H2020, No. 855005). M.H. was supported by the National Natural Science Foundation of China (42105160) and the Basic Research Fund of Chinese Academy of Meteorological Sciences (2023Z025).

## Author contributions
J.E.R. conceived the study, performed the analyses, and wrote, and revised the manuscript. T.W.C. and G.R.S. both conceived and supervised the study and wrote the manuscript. T.L. performed the analyses. S.S. provided major feedback on the analysis and interpretation. D.P., N.C., M.H. and S.P. all provided data and shaped the usage of the datasets, as well as interpretation of the results. C.M.Z. supervised and revised the manuscript. All authors edited and revised the manuscript.

## Competing interests
The authors declare no competing interests.
