## [Transparent Peer Review file · Communications Earth & Environment]

Forest edges are globally warmer than interiors and exceed optimal temperatures for vegetation productivity

Corresponding Author: Ms Josephine Reek

Version 0:

Decision Letter:

Dear Ms Reek,

Your manuscript titled "Temperatures at global forest edges are too high for optimal vegetation productivity" has now been seen by 3 reviewers, and we include their comments at the end of this message. They find your work of interest, but some important points are raised. We are interested in the possibility of publishing your study in Communications Earth & Environment, but would like to consider your responses to these concerns and assess a revised manuscript before we make a final decision on publication.

In revision, please address the following editorial thresholds:

- * Provide a compelling analysis of forest edge effects on local temperatures, fully considering potential factors modulating the effect.
- * Clarify the data selection and consider potential uncertainties in the analysis, providing robust results.

We therefore invite you to revise and resubmit your manuscript, along with a point-by-point response that takes into account the points raised. Please highlight all changes in the manuscript text file.

Please submit your point-by-point responses as a separate file, distinct from your cover letter where you can add responses to the Editors' comments that you do not want to be made available to the reviewers. Word files are preferred. We recommend that any figures, tables or graphs that are included in the response to reviewers are also included in the main article or Supplementary Information.

Please use the following link to submit your revised manuscript, point-by-point response to the referees' comments (which should be in a separate document to any cover letter), a tracked-changes version of the manuscript (as a PDF file) and the completed checklist:

Link Redacted

We hope to receive your revised paper within six weeks; please let us know if you aren't able to submit it within this time so that we can discuss how best to proceed. If we don't hear from you, and the revision process takes significantly longer, we may close your file. In this event, we will still be happy to reconsider your paper at a later date, as long as nothing similar has been accepted for publication at Communications Earth & Environment or published elsewhere in the meantime.

Please do not hesitate to contact us if you have any questions or would like to discuss these revisions further. We look forward to seeing the revised manuscript and thank you for the opportunity to review your work.

Best regards,

Mengjie Wang
Associate Editor
Communications Earth & Environment
@CommsEarth

EDITORIAL POLICIES AND FORMATTING

Editorial Policy: [Policy requirements](https://www.nature.com/documents/nr-editorial-policy-checklist.pdf) (Download the link to your computer as a PDF.)

- Behavioural and social science
- Ecological, evolutionary & environmental sciences
- Life sciences

<https://www.nature.com/documents/nr-reporting-summary.zip>

Furthermore, please align your manuscript with our format requirements, which are summarized on the following checklist: [Communications Earth & Environment formatting checklist](https://www.nature.com/documents/commsj-phys-style-formatting-checklist-article.pdf)

and also in our style and formatting guide [Communications Earth & Environment formatting guide](https://www.nature.com/documents/commsj-phys-style-formatting-guide-accept.pdf) .

*** DATA: Communications Earth & Environment endorses the principles of the Enabling FAIR data project (<http://www.copdess.org/enabling-fair-data-project/>). We ask authors to make the data that support their conclusions available in permanent, publically accessible data repositories. (Please contact the editor if you are unable to make your data available).

All Communications Earth & Environment manuscripts must include a section titled "Data Availability" at the end of the Methods section or main text (if no Methods). More information on this policy, is available at <http://www.nature.com/authors/policies/data/data-availability-statements-data-citations.pdf>.

If a community resource is unavailable, data can be submitted to generalist repositories such as [figshare](https://figshare.com/) or [Dryad Digital Repository](http://datadryad.org/). Please provide a unique identifier for the data (for example a DOI or a permanent URL) in the data availability statement, if possible. If the repository does not provide identifiers, we encourage authors to supply the search terms that will return the data. For data that have been obtained from publically available sources, please provide a URL and the specific data product name in the data availability statement. Data with a DOI should be further cited in the methods reference section.

REVIEWER COMMENTS:

Reviewer #1 (Remarks to the Author):

The manuscript "Temperatures at global forest edges are too high for optimal vegetation productivity" aims to assess the effects of forest fragmentation on local temperature. To do so, authors quantify temperature variation from forest edges to

interiors using satellite data. This global-scale analysis of edge effects is a novel and interesting aspect of the study, making it potentially relevant to a broad audience. The text is clear and succinct. However, despite these strengths, three main issues require attention:

1. Spatial scale and references missing: The main goal of the manuscript, as stated, is to assess the impact of forest fragmentation on local temperature. However, the authors quantify a local-scale phenomenon (edge effects) rather than forest fragmentation, which is a landscape-scale process. Two main concerns arise here. First, a previous study has already assessed the impacts of forest fragmentation on temperature using a landscape-scale approach (<https://link.springer.com/article/10.1007/s10980-020-01041-5>), which is not cited in the current manuscript. Secondly, that same study demonstrated that local-scale warming caused by edge effects cannot be extrapolated to predict higher landscape-level temperatures in more fragmented landscapes. This limitation has also been shown to apply to biodiversity patterns (DOI: 10.1111/conn.12992). Therefore, I believe the current manuscript should focus on edge effects only, rather than on forest fragmentation. Additionally, two reviews of forest fragmentation impacts on temperature are not referenced in the current manuscript (DOI: 10.1111/ele.12579 and DOI: 10.1007/s11284-016-1411-6).

2. Elevation: A major concern with the analysis is the absence of any control for elevation. In many montane landscapes, forest edges are often located at lower elevations than forest interiors, as human activity tends to be concentrated in lowland regions. In other words, the distance from the forest edge could be confounded with elevation. This is critical because temperature decreases with elevation, and thus, the conclusion that “forest edges worldwide are consistently warmer than interiors” might partially reflect differences in elevation. This potential confounding factor should be controlled for in the analysis and addressed explicitly in the Methods and/or Discussion.

3. Macroclimate temperature: Another central conclusion of the study is that “ $\Delta T/\Delta D$ is positively correlated with macroclimate temperature.” I believe this conclusion requires further evidence, as the R^2 values of the models are quite low (0.08 to 0.13). Given the large sample sizes, p-values are less informative here. Furthermore, the trend line in Fig. 2b does not seem to match the data points closely. The large scatter and low R^2 values suggest that some covariates, such as elevation (as noted earlier), may be missing from the models.

Reviewer #2 (Remarks to the Author):

This manuscript offers a global assessment of land surface temperature changes from forest edges, and the subsequent impact on vegetation productivity. The authors highlight differences between surface temperatures changes from edges across multiple biomes and seasons. There are many important scientific contributions associated with this paper, including the consequences of deforestation on changes in microclimates.

While the authors highlight the important changes associated with forest edges globally, there was a lack in explanation for why surface temperature was analyzed for only the year 2010. Was this a particularly hot year? Was it the year with the highest quality data? It was an El Niño year, which should be noted at some point in the manuscript. In addition, there were few details for the temporal overlap between the T_{surf} and the T_{surf} optima for ecosystem productivity. I recommend an additional paragraph either in the text or the supplementary material explaining these data, how they were derived, and why they were used in this analysis. Other details on the data should be added in the methods section, including reasoning for using those data and the uncertainties associated with them.

In addition, was there any qualification for how big a 'forest' needed to be? In other words, did the interior forest need to be a certain size for it to be classified as such?

Lastly, were there other biotic or abiotic variables accounted for within this analysis for the forests/points used? Any anthropogenic influences, droughts, defaunation, past/present fires, or other factors accounted for besides clouds and forest cover in the data filtering process? On this note, did the authors look into spatial autocorrelation issues in their analysis? If I missed it, please let me know. If not, please look into additional filtering processes to ensure quality data and analyses are used.

Overall, this paper provides important information on how forest fragmentation impacts land surface temperature as well as productivity.

Line by line comments:

28: replace 129,000,000 with 129 million

58: hotter environments is vague - what temperatures/which climates are you referring to?

68-72: rephrase sentences to make it clearer

81: how much hotter are the conditions?

Figures 1 and 3: Not intuitive to the reader because you have to read the caption to determine what line color corresponds with what season. Please add a legend to the figure for the line colors instead. Also, change the axes numbers to black and

check for axes number cutoffs (Figure3b)

106: How can you compare these datasets? Were both datasets from the same satellite? If not, what about the difference in timing of the data collection? Resolution of the data? Algorithms used to create the Tsurf measurements?

112: What is too high for Tsurf? Give specifics and what that means for productivity.

211: State what the actual product is you are using

212: Why did you use this product? Provide more detail.

212: Why did you choose the year 2010 and not a more recent year for studying LST? While the language from this paper represents recent findings, studying the temperatures from 14 years ago is quite a large gap compared to the present. 2010 was also an El Niño year, with higher than average temperatures. This information was never mentioned or acknowledged. Finally, the authors compared the LST from 2010 to FLUXNET data, but there is no mention of the timing of the two datasets and if there is any overlap. This is a critical component to this analysis.

215: Clearly state where the observed Tsurf data were collected, what year, and what kind of data they are in this summary paragraph

303: Removing 10% of the data to avoid outliers does not seem appropriate (given my knowledge of the analysis) without a solid justification. Do you have reasoning to believe that these data are true outliers? Please explain this in the MS or return the data to the analysis.

409-414: Move the R squared and p values higher in the plots to avoid the points to that is easier to read. Also, describe what the line is in the caption.

Reviewer #3 (Remarks to the Author):

This study calculated the temperature gradient from non-forest areas to interior forests. The authors found that forest edges exhibit higher temperatures than forest interiors, with this edge-warming effect being more pronounced in hotter environments. They also compared the observed temperature with the optimal temperature for forest productivity and found that the temperature in edge forests is too high for optimal vegetation productivity in summer. The authors processed extensive datasets to better understand temperature dynamics at forest edges, and their analyses are both important and interesting. The manuscript is well written, but a little bit short for discussion. It is great to see that they did a rigorous check on the data quality, especially the scene capture time and the validation using MODIS data (extended Fig. 4) and separating the dry and wet seasons in the tropics. I am overall positive about publishing this manuscript but several concerns regarding the methods and results that should be addressed before publication.

Major Comments:

1. The edge forest has been proven to grow worse in the tropical forest but grows better in the temperate forest (Smith et al., 2018; Morreale et al., 2021). The mechanisms underlying deforestation-induced cooling in boreal regions and warming in tropical regions are well understood. If the temperature edge effect operates similarly to the mechanism of deforestation, we would expect a cooling effect in temperate forests due to higher evapotranspiration (ET). However, your results contradict this expectation, although this point cannot be fully verified due to the coarse resolution of the current ET data. I suggest including a more detailed discussion of the mechanisms behind your findings. Overall, the discussion needs to be expanded. For example, what are the possible mechanisms or reasons for the finding of H2 and H3? Is the finding by Reinmann & Hutya (2017, PNAS) that edge effects enhanced forest growth consistent with your finding?
2. The same issue applies to your second hypothesis. Please provide reasonable explanations rather than merely describing the observed pattern (L92-94). Additionally, the significance of these results is unclear. The introduction is also too brief (L41-45), and I recommend expanding it to provide more context and clarity.
3. Comparing T30m with T500m may have some problems. At the forest edge, the Landsat LST signal (T30m) may include contributions from both forested and non-forested areas. As a result, even if the edge forest has the same temperature as the interior forest, T30m could still appear lower than T500m. 30 m and 500 m seem very arbitrary because the edge distance in some regions is much larger (see Bourgoin et al., 2024, Nature). A distance with a buffering zone or sensitivity tests with different distances would help clarify these issues.
4. DEM has consistently been considered in previous analyses (Li et al., 2015; Duveilier et al., 2018). While you argue that satellite-sensed temperature reflects the surface temperature of objects rather than the surrounding air, air temperature is still highly correlated with surface temperature (Hooker et al., 2018). Given the large area of each SxO (approximately 185 km x 180 km), there can be significant elevation variation, which should be accounted for in your study.

Minor Comments:

1. L8: The effects of forest fragmentation on local temperature in tropical regions have been well documented in a recent study (Zhu et al., 2023). This paper by Zhu et al. is very relevant to the manuscript, and I don't understand why it is not properly cited.

2. Several references are missing on L67-L72. Several studies have discussed the edge-scale impacts on temperature and other variables, such as biomass (Laurence et al., 1997; Broadbent et al., 2008; Zhao et al., 2021; Zhu et al., 2023).
3. Could you add a graph to illustrate your method for determining edge temperature?
4. I'm not sure if Figure 2b is necessary. Given the large number of data points, which are heavily overlapping, I would suggest presenting Figure 2 using a kernel density plot for better visualization.
5. In Extended Data Table 1, the dates for the wet and dry seasons are confusing. Why use data from 2012 for both seasons?
6. This study focuses on the year 2010, but Landsat 8 data only became available in 2013. Could you clarify how you used Landsat 8 data for this study?
7. Please provide units for $\Delta T/\Delta D$.
8. L300-301: Specify the maximum distance used when calculating $\Delta T/\Delta D$.
9. L301-302: Clarify what the beta coefficients represent. Are they the slopes?
10. Please provide your code for review.

References:

- Bourgoin, C., Ceccherini, G., Girardello, M. et al. Human degradation of tropical moist forests is greater than previously estimated. *Nature* 631, 570–576 (2024). <https://doi.org/10.1038/s41586-024-07629-0>
- Broadbent, E. N., Asner, G. P., Keller, M., Knapp, D. E., Oliveira, P. J. C., and Silva, J. N. (2008), Forest fragmentation and edge effects from deforestation and selective logging in the Brazilian Amazon, *Biological Conservation*, 141(7), 1745-1757.
- Duveiller, G., Hooker, J., and Cescatti, A. (2018), The mark of vegetation change on Earth's surface energy balance, *Nature Communications*, 9(1), 679.
- Hooker, J., Duveiller, G., and Cescatti, A. (2018), A global dataset of air temperature derived from satellite remote sensing and weather stations, *Scientific Data*, 5(1), 180246.
- Laurance, W. F., Laurance, S. G., Ferreira, L. V., Rankin-de Merona, J. M., Gascon, C., and Lovejoy, T. E. (1997), Biomass Collapse in Amazonian Forest Fragments, *Science*, 278(5340), 1117.
- Li, Y., Zhao, M., Motesharrei, S., Mu, Q., Kalnay, E., and Li, S. (2015), Local cooling and warming effects of forests based on satellite observations, *Nature Communications*, 6(1), 6603.
- Morreale, L. L., Thompson, J. R., Tang, X., Reinmann, A. B., and Hutyra, L. R. (2021), Elevated growth and biomass along temperate forest edges, *Nature Communications*, 12(1), 7181.
- Reinmann, A.B., Hutyra, L.R., Edge effects enhance carbon uptake and its vulnerability to climate change in temperate broadleaf forests, *Proc. Natl. Acad. Sci. U.S.A.* 114 (1) 107-112, <https://doi.org/10.1073/pnas.1612369114> (2017).
- Zhao, Z., Li, W., Ciais, P., Santoro, M., Cartus, O., Peng, S., Yin, Y., Yue, C., Yang, H., Yu, L., Zhu, L., and Wang, J. (2021), Fire enhances forest degradation within forest edge zones in Africa, *Nature Geoscience*, 14(7), 479-483.
- Zhu, L., Li, W., Ciais, P., He, J., Cescatti, A., Santoro, M., Tanaka, K., Cartus, O., Zhao, Z., Xu, Y., Sun, M., and Wang, J. (2023), Comparable biophysical and biogeochemical feedbacks on warming from tropical moist forest degradation, *Nature Geoscience*, 16(3), 244-249.

Communications Earth & Environment is committed to improving transparency in authorship. As part of our efforts in this direction, we are now requesting that all authors identified as 'corresponding author' create and link their Open Researcher and Contributor Identifier (ORCID) with their account on the Manuscript Tracking System prior to acceptance. ORCID helps the scientific community achieve unambiguous attribution of all scholarly contributions. You can create and link your ORCID from the home page of the Manuscript Tracking System by clicking on 'Modify my Springer Nature account' and following the instructions in the link below. Please also inform all co-authors that they can add their ORCIDs to their accounts and that they must do so prior to acceptance.

If you experience problems in linking your ORCID, please contact the Platform Support Helpdesk.

Version 1:

Decision Letter:

Dear Ms Reek,

Your manuscript titled "Forest edges are globally warmer than interiors and increasingly exceed thermal thresholds for optimal vegetation productivity" has now been seen by our reviewers, whose comments appear below. In light of their advice we are delighted to say that we are happy, in principle, to publish a suitably revised version in Communications Earth & Environment.

We therefore invite you to revise your paper one last time to address the remaining concerns of our reviewers. At the same time we ask that you edit your manuscript to comply with our format requirements and to maximise the accessibility and therefore the impact of your work.

EDITORIAL REQUESTS:

****Please take care to match our formatting and policy requirements. We will check revised manuscript and return manuscripts that do not comply. Such requests will lead to delays. ****

SUBMISSION INFORMATION:

OPEN ACCESS:

Communications Earth & Environment is a fully open access journal. Articles are made freely accessible on publication. For further information about article processing charges, open access funding, and advice and support from Nature Research, please visit <https://www.nature.com/commsenv/open-access>

Link Redacted

Best regards,

Mengjie Wang

Associate Editor, Communications Earth & Environment

<https://www.nature.com/commsenv/>

Consulting Editor, Communications Sustainability

<https://www.nature.com/commssustain/>

Bluesky: @commsearth.nature.com; @commssustain.nature.com

REVIEWERS' COMMENTS:

Reviewer #2 (Remarks to the Author):

The authors have submitted a revised version of their manuscript titled 'Forest edges are globally warmer than interiors and increasingly exceed thermal thresholds for optimal vegetation productivity', with a substantial amount of changes made after the first round of review. The latest version provides more detailed information on the methods used and additional analyses to back their conclusions. The inclusion of elevation and accounting of spatial autocorrelation strengthen the findings, and I applaud the authors efforts. The figures have also been adjusted to clarify the results for the reader. The publicly available code is also a helpful and necessary addition. The latest manuscript is more robust and provides insightful conclusions to

the field. No additional changes are requested.

Reviewer #3 (Remarks to the Author):

I appreciate the authors' efforts, and my concerns have been addressed properly.

REVIEWER COMMENTS:

Reviewer #1 (Remarks to the Author):

The manuscript “Temperatures at global forest edges are too high for optimal vegetation
productivity” aims to assess the effects of forest fragmentation on local temperature. To do so,
authors quantify temperature variation from forest edges to interiors using satellite data. This
global-scale analysis of edge effects is a novel and interesting aspect of the study, making it
potentially relevant to a broad audience. The text is clear and succinct. However, despite these
strengths, three main issues require attention:

**Reply 1:** We thank the reviewer for the valuable feedback on our manuscript. We have
carefully considered the points and adjusted the manuscript accordingly. Thus, we now make
a clearer distinction between fragmentation and edge effects. We also improved our models
by including elevation and expanded their discussion to acknowledge additional factors that
may influence our findings.

We appreciate the points made and believe that the adjustments have greatly enhanced the
manuscript by improving the models and adding precision to the discussion.

1. Spatial scale and references missing: The main goal of the manuscript, as stated, is to
assess the impact of forest fragmentation on local temperature. However, the authors quantify
a local-scale phenomenon (edge effects) rather than forest fragmentation, which is a
landscape-scale process. Two main concerns arise here. First, a previous study has already
assessed the impacts of forest fragmentation on temperature using a landscape-scale
approach (<https://link.springer.com/article/10.1007/s10980-020-01041-5>), which is not cited in
the current manuscript. Secondly, that same study demonstrated that local-scale warming
caused by edge effects cannot be extrapolated to predict higher landscape-level temperatures
in more fragmented landscapes. This limitation has also been shown to apply to biodiversity
patterns (DOI: 10.1111/conl.12992). Therefore, I believe the current manuscript should focus
on edge effects only, rather than on forest fragmentation. Additionally, two reviews of forest
fragmentation impacts on temperature are not referenced in the current manuscript (DOI:
10.1111/ele.12579 and DOI: 10.1007/s11284-016-1411-6).

**Reply 2:** We thank the reviewer for making this very important point. We have added a
discussion on the fact that temperature edge effects cannot be scaled up to landscape level
temperatures, as well as the limitation of these effects to just the forested parts of a fragmented
landscape (lines 33-42). We have also taken care to only refer to fragmentation as a process
that can generate edges, and added a sentence to clarify that this study investigates the edge
effects specifically (lines 39-42):

*“While fragmentation changes the amount of edge areas, this study is specifically looking at*
*edge effects – the microclimatic differences between forest edges and interiors – which cannot*
*be directly extrapolated to explain temperature patterns across entire fragmented*
*landscapes.”*

We were also happy to include and cite the additional references which have improved the
presentation of the topic (lines 29, 30, 34, 35, 38, 106).

2. Elevation: A major concern with the analysis is the absence of any control for elevation. In
many montane landscapes, forest edges are often located at lower elevations than forest
interiors, as human activity tends to be concentrated in lowland regions. In other words, the
distance from the forest edge could be confounded with elevation. This is critical because
temperature decreases with elevation, and thus, the conclusion that “forest edges worldwide
are consistently warmer than interiors” might partially reflect differences in elevation. This
potential confounding factor should be controlled for in the analysis and addressed explicitly
in the Methods and/or Discussion.

Reply 3: We appreciate the reviewer pointing this out and refer also to Reply 26. We have now included control for elevation in all models and explained this in the methods (Methods, esp. lines 270-274 and 278-282). Our results and conclusions remain largely unchanged, and all Figures and analyses presented now account for this.

3. Macroclimate temperature: Another central conclusion of the study is that “ $\Delta T/\Delta D$ is positively correlated with macroclimate temperature.” I believe this conclusion requires further evidence, as the R^2 values of the models are quite low (0.08 to 0.13). Given the large sample sizes, p-values are less informative here. Furthermore, the trend line in Fig. 2b does not seem to match the data points closely. The large scatter and low R^2 values suggest that some covariates, such as elevation (as noted earlier), may be missing from the models.

Reply 4: We thank the reviewer for making this point and agree that there are certainly many other influential factors apart from macroclimatic temperature. We now clarify this in the text in lines 104-108.

Additionally, with the models that were updated according to the reviewer’s suggestion (Reply 3), the R^2 values are now somewhat higher (0.197 and 0.163 for the main models, Figure 2). We have also chosen a quadratic trend line and made the visualisation of the datapoint density clearer. Altogether, we believe that these updates greatly improve the analysis.

Reviewer #2 (Remarks to the Author):

This manuscript offers a global assessment of land surface temperature changes from forest edges, and the subsequent impact on vegetation productivity. The authors highlight differences between surface temperatures changes from edges across multiple biomes and seasons. There are many important scientific contributions associated with this paper, including the consequences of deforestation on changes in microclimates.

Reply 5: We thank the reviewer for the thoughtful feedback and have carefully considered the specific points made below. Thus, we now explain and discuss the datasets in more detail, and also added additional analyses to ensure the robustness of our findings. We believe that this has greatly improved the manuscript overall and discuss individual points in more detail below.

While the authors highlight the important changes associated with forest edges globally, there was a lack in explanation for why surface temperature was analyzed for only the year 2010. Was this a particularly hot year? Was it the year with the highest quality data? It was an El Niño year, which should be noted at some point in the manuscript. In addition, there were few details for the temporal overlap between the Tsurf and the Tsurf optima for ecosystem productivity. I recommend an additional paragraph either in the text or the supplementary material explaining these data, how they were derived, and why they were used in this analysis. Other details on the data should be added in the methods section, including reasoning for using those data and the uncertainties associated with them.

Reply 6: We thank the reviewer for pointing out these gaps, which also fit Reply 18. We have now expanded the methods to provide details on the points above. Specifically:

Choice of the year (lines 201-202):

“The year 2010 was used for this analysis because it allowed us to use the Landsat-derived Tsurf data as well as high quality and high resolution forest cover data.”

El Niño (lines 254-257):

“Our study year of 2010 was an El Niño year, with relatively high temperatures. [...]”

Temporal overlap between datasets (line 344-345):
*“T_{opt} could thus be calculated using data from 2003 – 2013, which includes our study year*
*2010.”*

More details and discussion on the datasets (Methods, esp. lines 207-210, 338-355).

In addition, was there any qualification for how big a 'forest' needed to be? In other words, did
the interior forest need to be a certain size for it to be classified as such?

**Reply 7:** We have not specified a minimum size requirement. “Forest” is just defined as a pixel
with at least 30% canopy cover (line 221). Thus, the only size requirement is that there is a
large enough area to lend the 30% canopy cover to a 30x30 meter pixel.

Lastly, were there other biotic or abiotic variables accounted for within this analysis for the
forests/points used? Any anthropogenic influences, droughts, defaunation, past/present fires,
or other factors accounted for besides clouds and forest cover in the data filtering process?
On this note, did the authors look into spatial autocorrelation issues in their analysis? If I
missed it, please let me know. If not, please look into additional filtering processes to ensure
quality data and analyses are used.

**Reply 8:** We appreciate the reviewer bringing up these important points. The datapoints used
were just filtered to be within 1000m of an edge and on land on cloud free days, and, as
pointed out by the reviewer, forest is defined as 30% or more canopy cover (Reply 7). During
the modelling step, we control for satellite scene and overpass, as well as elevation. We are
using published datasets as they are described in their publications. Thus, we trust the original
authors to have handled quality control as described and do not filter them further. However,
we have expanded description of these datasets (Methods, esp. lines 207-210, 338-355).

We did not filter for any anthropogenic or other influences that led to the creation of the edges,
as we were mostly interested in forest edges as a landscape feature, irrespective of their
origin. However, this would be very interesting, and a nice follow-up, which we have now
specifically noted in the text (line 107-109):

*“Moreover, the origin and anthropogenic modifications of forest edges may influence*
*temperature dynamics, presenting a valuable avenue for future research.”*

With regards to autocorrelation, we controlled for the satellite scenes, which controls for all
T_{surf}-affecting variation (abiotic and otherwise) that makes a focal SxO different from another
SxO, leaving us with T_{surf} variation attributable to distance from forest edge. To ensure that
our SxO correction is sufficient, and the results are not based on autocorrelation, we have
added a Moran’s I calculation on the residuals of the BAMs (Supplementary Table 6, lines
307-311). While the values were significant, they are very low (0.0268 for boreal winter and
<0.01 for all other models), meaning there is very little residual autocorrelation. Thus, we do
not expect our conclusions to be influenced by it.

Overall, this paper provides important information on how forest fragmentation impacts land
surface temperature as well as productivity.

Line by line comments:

28: replace 129,000,000 with 129 million

**Reply 9:** Done (line 24)

58: hotter environments is vague - what temperatures/which climates are you referring to?

**Reply 10:** We thank the reviewer for pointing out the unclarity. We were not referring to a
specific cut-off, but to a continuous change with increasing temperature. This has now been
clarified (line 66).

68-72: rephrase sentences to make it clearer

**Reply 11:** This has been reformulated for clarity (lines 82-86).

81: how much hotter are the conditions?

**Reply 12:** We appreciate the reviewer pointing out the unclarity. While two arbitrary points of
“heat conditions” could be compared, we are not focussed on specific values. The sentence
has been reformulated to reflect the nature of the positive correlation and continuous
relationship (lines 93-94, Reply 10).

Figures 1 and 3: Not intuitive to the reader because you have to read the caption to determine
what line color corresponds with what season. Please add a legend to the figure for the line
colors instead. Also, change the axes numbers to black and check for axes number cutoffs
(Figure3b)

**Reply 13:** We thank the reviewer for pointing these out. We have updated the figures
accordingly and believe that it certainly improves interpretability. (Figures 1 and 3)

106: How can you compare these datasets? Were both datasets from the same satellite? If
not, what about the difference in timing of the data collection? Resolution of the data?
Algorithms used to create the T_{surf} measurements?

**Reply 14:** We appreciate the reviewer pointing out the missing explanation and have added
a section discussing and comparing the datasets (lines 346-355). While the datasets are not
based on the same satellite, they both measured the same property, namely surface
temperature. The T_{opt} dataset was averaged over several years (2003 – 2013). This provides
stability to the estimate and also includes our study year of 2010. Additional information on
these datasets is now provided in the methods section (lines 207-210, 338-355).

112: What is too high for T_{surf}? Give specifics and what that means for productivity.

**Reply 15:** We thank the reviewer for pointing out the unclarity and have expanded the
explanation. On what “too high” means, as well as what this means for productivity. lines 137-
140:

*“In the tropics, both interior and edge T_{surf} exceeded the optimal temperature for productivity,
in line with previous findings based on air temperature³⁰. Ecosystem-level productivity is thus
expected to be lower than it would under cooler, optimal temperature conditions, especially in
the hotter edge areas.”*

211: State what the actual product is you are using

**Reply 16:** We have stated the dataset specification more clearly (lines 191-192).

212: Why did you use this product? Provide more detail.

**Reply 17:** We have added more detail on the specific choice of the product (lines 203-207),
as well as on the product itself (lines 207-210).

212: Why did you choose the year 2010 and not a more recent year for studying LST? While

the language from this paper represents recent findings, studying the temperatures from 14
222 years ago is quite a large gap compared to the present. 2010 was also an El Nino year, with
223 higher than average temperatures. This information was never mentioned or acknowledged.
Finally, the authors compared the LST from 2010 to FLUXNET data, but there is no mention
of the timing of the two datasets and if there is any overlap. This is a critical component to this
analysis.

**Reply 18:** We thank the reviewer for pointing out these gaps in the explanation and also point
to Reply 6. We have explained our rationale for the choice of study period (lines 201-202),
including El Nino (lines 254-257). While our data are not directly compared to FLUXNET data,
it was indeed used for verification of the T_{opt} dataset that we compare to. We acknowledge
that this was rather unclear in the previous version, and have expanded the explanations and
discussions of the datasets used (lines 207-210, 338-355) as well as their temporal overlap
(lines 346-355).

215: Clearly state where the observed T_{surf} data were collected, what year, and what kind of
data they are in this summary paragraph

**Reply 19:** We have added these specifications to the summary paragraph (lines 192-194).

303: Removing 10% of the data to avoid outliers does not seem appropriate (given my
knowledge of the analysis) without a solid justification. Do you have reasoning to believe that
these data are true outliers? Please explain this in the MS or return the data to the analysis.

**Reply 20:** For the main analysis we are now using a Hampel filter to identify outliers (median
+/- 3 mean absolute deviations), which is a standard method. However, we now also provide
the analysis without any outlier removal in the supplementary information (Supplementary
Figure 3). Our findings are robust to either procedure, however, based on the plots provided
in that supplementary analysis, we decided to use an outlier filter for the main analysis. Should
the reviewer have any suggestion for an improved technique/ handling of this, we are also
open to that. (lines 320-323)

409-414: Move the R squared and p values higher in the plots to avoid the points to that is is
easier to read. Also, describe what the line is in the caption.

**Reply 21:** The R^2 value has been moved up (above the points), the p-values are now in the
Figure legend (Figure 2). We appreciate the suggestion and improved readability.

Reviewer #3 (Remarks to the Author):

This study calculated the temperature gradient from non-forest areas to interior forests. The
authors found that forest edges exhibit higher temperatures than forest interiors, with this
edge-warming effect being more pronounced in hotter environments. They also compared the
observed temperature with the optimal temperature for forest productivity and found that the
temperature in edge forests is too high for optimal vegetation productivity in summer. The
authors processed extensive datasets to better understand temperature dynamics at forest
edges, and their analyses are both important and interesting. The manuscript is well written,
but a little bit short for discussion. It is great to see that they did a rigorous check on the data
quality, especially the scene capture time and the validation using MODIS data (extended Fig.
4) and separating the dry and wet seasons in the tropics. I am overall positive about publishing
this manuscript but several concerns regarding the methods and results that should be
addressed before publication.

**Reply 22:** We sincerely appreciate the reviewer's thoughtful and positive feedback on our
manuscript. We also thank them for the suggestions for improvements and have carefully
considered the individual points. Thus, we have expanded the discussion and provide more
information on possible mechanisms behind our findings. We have also updated the models
to control for elevation and included supplementary analyses to ensure robustness.

Overall, we believe that these edits have greatly enhanced the manuscript by improving the
models, as well as clarity and context.

Major Comments:

1. The edge forest has been proven to grow worse in the tropical forest but grows better in the
temperate forest (Smith et al., 2018; Morreale et al., 2021). The mechanisms underlying
deforestation-induced cooling in boreal regions and warming in tropical regions are well
understood. If the temperature edge effect operates similarly to the mechanism of
deforestation, we would expect a cooling effect in temperate forests due to higher
evapotranspiration (ET). However, your results contradict this expectation, although this point
cannot be fully verified due to the coarse resolution of the current ET data. I suggest including
a more detailed discussion of the mechanisms behind your findings. Overall, the discussion
needs to be expanded. For example, what are the possible mechanisms or reasons for the
finding of H2 and H3? Is the finding by Reinmann & Hutyra (2017, PNAS) that edge effects
enhanced forest growth consistent with your finding?

**Reply 23:** We appreciate the reviewer's point and have expanded the discussion to include
more discussion on the possible mechanisms as well as a specific discussion of the Reinmann
& Hutyra (2017, PNAS) publication, which is indeed very relevant. (lines 112-123, 147-163)

2. The same issue applies to your second hypothesis. Please provide reasonable explanations
rather than merely describing the observed pattern (L92-94). Additionally, the significance of
these results is unclear. The introduction is also too brief (L41-45), and I recommend
expanding it to provide more context and clarity.

**Reply 24:** We have expanded the discussion (e.g. 112-123) as well as the introduction (e.g.
33-42, 54-56) to suggest mechanisms as well as better explain the significance of these
effects. This includes patterns in albedo and evapotranspirative cooling, as well as implications
regarding edge effects and fragmentation.

3. Comparing T30m with T500m may have some problems. At the forest edge, the Landsat
LST signal (T30m) may include contributions from both forested and non-forested areas. As
a result, even if the edge forest has the same temperature as the interior forest, T30m could
still appear lower than T500m. 30 m and 500 m seem very arbitrary because the edge distance
in some regions is much larger (see Bourgoïn et al., 2024, Nature). A distance with a buffering
zone or sensitivity tests with different distances would help clarify these issues.

**Reply 25:** We chose 30m for the edge to avoid the influence of non-forest parts, as 30m is
our pixel resolution and we measure Euclidian distances to the forest edge. Thus, even when
a point falls exactly on those 30m from an edge, the temperature is measured on a 30m grid
and thus not further away than the 30m to the edge. 500m were chosen because we see in
F1 that the curves have flattened out by then and temperatures appear stable. However, the
reviewer makes an important point about a degree of arbitrariness in these choices, and we
have also added a supplementary analysis using 50m and 1km instead. (Supplementary
Figures 2 and 4, lines 304-306, 377-379)

4. DEM has consistently been considered in previous analyses (Li et al., 2015; Duveillier et al.,
2018). While you argue that satellite-sensed temperature reflects the surface temperature of

objects rather than the surrounding air, air temperature is still highly correlated with surface
temperature (Hooker et al., 2018). Given the large area of each SxO (approximately 185 km
x 180 km), there can be significant elevation variation, which should be accounted for in your
study.

**Reply 26:** We appreciate the reviewer making this important point and also refer to Reply 3.
We now control for elevation in all our models (Methods, esp. lines 270-274 and 278-282). We
have observed no substantial change to our results and present only elevation-controlled
models in our updated manuscript.

Minor Comments:

1. L8: The effects of forest fragmentation on local temperature in tropical regions have been
well documented in a recent study (Zhu et al., 2023). This paper by Zhu et al. is very relevant
to the manuscript, and I don't understand why it is not properly cited.

**Reply 27:** We thank the reviewer for drawing our attention to this, and are now referencing
the study (line 33).

2. Several references are missing on L67-L72. Several studies have discussed the edge-scale
impacts on temperature and other variables, such as biomass (Laurence et al., 1997;
Broadbent et al., 2008; Zhao et al., 2021; Zhu et al., 2023).

**Reply 28:** We thank the reviewer for pointing us to those references, and have added them
(lines 30, 33, 148)

3. Could you add a graph to illustrate your method for determining edge temperature?

**Reply 29:** We have expanded the explanation for determining edge temperatures (lines 296-
298).

4. I'm not sure if Figure 2b is necessary. Given the large number of data points, which are
heavily overlapping, I would suggest presenting Figure 2 using a kernel density plot for better
visualization.

**Reply 30:** We appreciate the suggestion and are now using density plots for Figure 2 (Figure
2).

5. In Extended Data Table 1, the dates for the wet and dry seasons are confusing. Why use
data from 2012 for both seasons?

**Reply 31:** We greatly appreciate the catch, this was a mistake. Wet and dry season also use
2010 data, the same as the rest of the analyses (Extended Data Table 1).

6. This study focuses on the year 2010, but Landsat 8 data only became available in 2013.
Could you clarify how you used Landsat 8 data for this study?

**Reply 32:** We thank the reviewer for pointing out the incongruency. We were originally
referring to the fact that the published dataset is available for those satellites, but based on
the timeframe, our analysis uses data from Landsat 5. We acknowledge that this was unclear
and have now clarified it (line 201).

7. Please provide units for $\Delta T/\Delta D$.

**Reply 33:** We have now added those units ($-\Delta T/\Delta D \rightarrow -(^{\circ}\text{C}/\log_{10}(\text{m}))$). (Figure 2)

8. L300-301: Specify the maximum distance used when calculating $\Delta T/\Delta D$.

**Reply 34:** Done (line 317)

9. L301-302: Clarify what the beta coefficients represent. Are they the slopes?

**Reply 35:** Yes, they are, we have clarified this now (lines 319-320).

10. Please provide your code for review.

Reply 36: Data and code can be found under <https://figshare.com/s/4ca727565ccc924dc77a>.

References:

- Bourgoin, C., Ceccherini, G., Girardello, M. et al. Human degradation of tropical moist forests is greater than previously estimated. *Nature* 631, 570–576 (2024). <https://doi.org/10.1038/s41586-024-07629-0>
- Broadbent, E. N., Asner, G. P., Keller, M., Knapp, D. E., Oliveira, P. J. C., and Silva, J. N. (2008), Forest fragmentation and edge effects from deforestation and selective logging in the Brazilian Amazon, *Biological Conservation*, 141(7), 1745-1757.
- Duveiller, G., Hooker, J., and Cescatti, A. (2018), The mark of vegetation change on Earth's surface energy balance, *Nature Communications*, 9(1), 679.
- Hooker, J., Duveiller, G., and Cescatti, A. (2018), A global dataset of air temperature derived from satellite remote sensing and weather stations, *Scientific Data*, 5(1), 180246.
- Laurance, W. F., Laurance, S. G., Ferreira, L. V., Rankin-de Merona, J. M., Gascon, C., and Lovejoy, T. E. (1997), Biomass Collapse in Amazonian Forest Fragments, *Science*, 278(5340), 1117.
- Li, Y., Zhao, M., Motesharrei, S., Mu, Q., Kalnay, E., and Li, S. (2015), Local cooling and warming effects of forests based on satellite observations, *Nature Communications*, 6(1), 6603.
- Morreale, L. L., Thompson, J. R., Tang, X., Reinmann, A. B., and Hutyrá, L. R. (2021), Elevated growth and biomass along temperate forest edges, *Nature Communications*, 12(1), 7181.
- Reinmann, A.B., Hutyrá, L.R., Edge effects enhance carbon uptake and its vulnerability to climate change in temperate broadleaf forests, *Proc. Natl. Acad. Sci. U.S.A.* 114 (1) 107-112, <https://doi.org/10.1073/pnas.1612369114> (2017).
- Zhao, Z., Li, W., Ciais, P., Santoro, M., Cartus, O., Peng, S., Yin, Y., Yue, C., Yang, H., Yu, L., Zhu, L., and Wang, J. (2021), Fire enhances forest degradation within forest edge zones in Africa, *Nature Geoscience*, 14(7), 479-483.
- Zhu, L., Li, W., Ciais, P., He, J., Cescatti, A., Santoro, M., Tanaka, K., Cartus, O., Zhao, Z., Xu, Y., Sun, M., and Wang, J. (2023), Comparable biophysical and biogeochemical feedbacks on warming from tropical moist forest degradation, *Nature Geoscience*, 16(3), 244-249.